# How Does Generative Retrieval Scale to Millions of Passages?

**Ronak Pradeep**[*][†][§]**, Kai Hui**[*]**, Jai Gupta, Adam D. Lelkes, Honglei Zhuang**
**Jimmy Lin**[§]**, Donald Metzler, Vinh Q. Tran**[*]
Google Research, [§]University of Waterloo
rpradeep@uwaterloo.ca, {kaihuibj,vqtran}@google.com

## Abstract

The emerging paradigm of generative retrieval re-frames the classic information retrieval problem into a sequence-to-sequence modeling task, forgoing external indices and encoding an entire document corpus within a single Transformer. Although many different approaches have been proposed to improve the effectiveness of generative retrieval, they have only been evaluated on document corpora on the order of 100K in size. We conduct the first empirical study of generative retrieval techniques across various corpus scales, ultimately scaling up to the entire MS MARCO passage ranking task with a corpus of 8.8M passages and evaluating model sizes up to 11B parameters. We uncover several findings about scaling generative retrieval to millions of passages; notably, the central importance of using synthetic queries as document representations during indexing, the ineffectiveness of existing proposed architecture modifications when accounting for compute cost, and the limits of naively scaling model parameters with respect to retrieval performance. While we find that generative retrieval is competitive with state-of-the-art dual encoders on small corpora, scaling to millions of passages remains an important and unsolved challenge. We believe these findings will be valuable for the community to clarify the current state of generative retrieval, highlight the unique challenges, and inspire new research directions.

## 1 Introduction

For the last several years, dual encoders (Gillick et al., 2018; Karpukhin et al., 2020; Ni et al., 2022b; Chen et al., 2022) have dominated the landscape for first-stage information retrieval. They model relevance by mapping queries and documents into the same embedding space, optimized via contrastive learning (Hadsell et al., 2006; Gao et al.,

2021). Dense embeddings are pre-computed for all documents in a corpus and stored in an external index, enabling fast approximate nearest neighbor search (Vanderkam et al., 2013; Johnson et al., 2021) to retrieve relevant documents. Cross-encoders based on large Transformer models (Nogueira and Cho, 2019; Nogueira et al., 2020; Pradeep et al., 2021b) often function on top of these retrieved documents to further refine the top results.

Recently, the emerging paradigm of generative retrieval (De Cao et al., 2020; Tay et al., 2022) sought to replace this entire process with a single sequence-to-sequence Transformer model (Sutskever et al., 2014; Vaswani et al., 2017), showing promising results against dual encoders given a sufficiently small corpus size. Since then, various techniques, such as (Zhuang et al., 2022b; Bevilacqua et al., 2022; Zhou et al., 2022; Wang et al., 2022; Chen et al., 2023), have aimed to improve the effectiveness of generative retrieval models, either with alternative document identifier formulations, architecture changes, or training objectives. Such work, however, has only evaluated generative retrieval over relatively small corpora on the order of 100K documents, such as Natural Questions (Kwiatkowski et al., 2019), TriviaQA (Joshi et al., 2017), or small subsets of the MS MARCO document ranking task (Nguyen et al., 2016). Despite these research contributions, several open questions remain unanswered, including *how well* current generative retrieval techniques work on larger corpora and *which aspects* of generative retrieval models proposed so far are vital as we scale the corpora.

In this paper, we conduct the first empirical study of generative retrieval techniques over the entire MS MARCO passage-level corpus, evaluating its effectiveness over 8.8M passages. We select popular approaches in recent works and evaluate them first on Natural Questions and TriviaQA to establish a definitive ablation of techniques in a con-

---

[*]Equal Contribution.
[†]Work completed while a Student Researcher at Google.

trolled setup. Our experiments mainly focus on evaluating techniques proposed by Tay et al. (2022), Zhuang et al. (2022b), and Wang et al. (2022). We evaluate three main design aspects in our study:

- The impact of target-space document identifiers (DocIDs) exploring Atomic, Naive, Semantic, and 2D Semantic DocIDs.

- Document representation, assessing the use of document tokens, ground truth queries, and synthetic queries, as per Nogueira and Lin (2019).

- Model design techniques including prefix-aware weight-adaptive (PAWA) decoding, constrained decoding, and consistency loss.

We then scale up the corpus size leveraging the MS MARCO passage ranking task, beginning with a subset of 100K passages before increasing the count to 1M and 8.8M passages (the entire set). Incrementally doing so allows us to establish which techniques remain effective as corpus size and difficulty scale.

Finally, to explore the effect of model scaling on retrieval effectiveness on large corpora, we evaluate a set of techniques with promising results at the T5.1.1-Base (220M) scale (Raffel et al., 2020) and modify the parameterization to consider up to 11B parameters. As the parameter distributions vary between methods, for instance, Atomic DocIDs result in additional parameters equal to the embedding dimension times corpus size, while the Naive DocIDs do not cost anything beyond the core Transformer model, we aim to provide some insight into the trade-off of different parameter allocations on a large corpus.

While our experimental findings are nuanced, we summarize the main findings as follows:

1. Of the methods considered, we find synthetic query generation to be the single most critical component as corpus size grows (other than model scaling.) That is, defining the task of generative retrieval as solely mapping from synthetic queries to DocIDs is the most effective modeling strategy, with others largely unnecessary.

2. As corpus size increases, discussion of compute cost is crucial. Methods that expand the parameter count beyond what is used by the original pre-trained T5 checkpoint are more effective. We find this crucial detail unstated in prior work like Wang et al. (2022). However, the effectiveness improvements vanish as we scale up the naive approach to similar parameter sizes. Following Dehghani et al. (2022), we note that the parameter count is not the entire story and provide more discussion regarding model comparisons and trade-offs in Section A.3.2.

3. Increasing the model size is necessary for improved generative retrieval effectiveness. However, for the best Sequential DocIDs, effectiveness does not improve past a certain point — peaking at XL (3B) with a slightly worse score using XXL (11B) under fixed experimental settings. We find this counter-intuitive to the common conception that model capacity limits the success of generative retrieval in prior studies of generative retrieval.

In the MS MARCO passage ranking task, we found that a scaled model, trained only on synthetic queries and leveraging Naive DocIDs, outperforms all other techniques we considered. On a small subset of 100K passages, a T5-Base model trained with this strategy achieves 82.4 MRR@10 (Section A.3.1), competitive with GTR-Base (Ni et al., 2022b) at 83.2 MRR@10. However, on the setting with 8.8M passages, a T5-XL model trained with this approach achieves only 26.7 MRR@10.

The field of generative retrieval is rapidly evolving, yet matching the effectiveness of top-tier dense retrieval models at scale remains a significant, unresolved challenge. Our results suggest the need for continued research into generative retrieval and more fundamental advances to the paradigm before we can fully leverage the power of scaling up model parameters. We believe our findings will help the research community better understand the challenges faced while applying generative retrieval models to larger corpora and inspire new research in this field.

## 2 Methods

We revisit the design details of the generative retrieval method, using the Differentiable Search Index (DSI) (Tay et al., 2022) as the baseline, and describe multiple techniques introduced in subsequent works that we ablate and study in this work. For a comprehensive review of related work, we refer the reader to Section A.1 in the Appendix.

## 2.1 Background

DSI (Tay et al., 2022) transforms the retrieval task into sequence-to-sequence (seq2seq) modeling, using queries as inputs and relevant document identifiers (DocIDs) as generation targets. The corpus, namely the mapping between the document's content and its identifier, is encoded using the parameters of the LLM. DSI achieves this by leveraging two seq2seq tasks: indexing and retrieval. During training, the model learns to generate the DocID given the document content (indexing task) or a relevant query (retrieval task). At inference, the model processes a query and generates a ranked list of identifiers as retrieval results.

## 2.2 Inputs and Targets

In the framework discussed, DSI learns to encode the mapping between the long-form textual representation of a document and its identifier in its parameters while also learning to fetch the same identifier when it receives a relevant query as input.

Two crucial design choices are how documents are represented (i.e., the inputs in the indexing task) and how document identifiers (DocIDs) are represented (i.e., the targets in both indexing and retrieval tasks). One primary consideration is the challenge of encoding long textual sequences with a Transformer (Vaswani et al., 2017)-based LLM, making it hard to index entire documents. Another is that naive identifiers from existing datasets could be sub-optimal due to their lack of semantic meaning.

### 2.2.1 Document Representations

One straightforward idea is to pick a text span from the document as a representation. DSI considers the first 64 tokens (FirstP) in each document, whereas Wang et al. (2022) leverages ten randomly selected chunks of 64 consecutive tokens, a technique they call Document As Query (DaQ). We evaluate the FirstP and DaQ approaches individually and in conjunction when working with lengthy documents in Natural Questions and TriviaQA datasets. In the case of MS MARCO, which has short passages, FirstP and DaQ are essentially the same, assuming sufficient context length.

### 2.2.2 Synthetic Query Generation

In training the model for retrieval tasks, the natural baseline uses existing labeled data, i.e., queries from the retrieval dataset as inputs and the DocIDs labeled as relevant as targets. However, as argued in Zhuang et al. (2022b) and Wang et al. (2022), there are two kinds of gaps between the index and retrieval tasks. First is the data distribution gap: queries for the retrieval task are short and request specific information, while the documents for the indexing task are long and convey information. Second is the coverage gap: the model is exposed to the entire corpus during the training of the indexing task, while only positive examples have associated queries in the retrieval task. The latter problem is exacerbated in the MS MARCO passage ranking task as only 550K passages have an associated query for training the retrieval task, while the indexing task has to learn to encode all 8.8M passages in the corpus.

To mitigate this gap, they propose generating synthetic queries for each document using models like docT5query (Nogueira and Lin, 2019). They then trained a generative retrieval model to predict the DocIDs given the corresponding synthetic queries. We can also think of these synthetic queries as alternative document representations.

### 2.2.3 Document Identifiers

In this work, we consider four kinds of different identifiers: the three kinds of document identifiers from DSI (Tay et al., 2022): unstructured atomic identifiers (Atomic DocIDs), naive string identifiers (Naive DocIDs), semantically structured identifiers (Semantic DocIDs), and the 2D Semantic DocIDs from Wang et al. (2022).

**Atomic DocIDs.** We treat each DocID as a single, or "atomic" token in this setting. The decoder, then, only needs to run for a single decoding step; we then sort the logits of the DocIDs to obtain the ranked document list. Here, each document requires an additional unique token in the model vocabulary. As a result, the model's parameters increase by the corpus size times the embedding dimension, becoming potentially expensive for large corpora. When considering millions of documents, we apply two optimizations to make the implementation more feasible. First, the embedding table of the encoder is adjusted to include only the standard T5 vocabulary, while the output projection of the decoder corresponds to DocIDs. Second, we take special care to ensure the output projection is properly sharded across cores to distribute memory cost to allow scaling. We achieve this by setting partitioning rules in the t5x framework (Roberts et al., 2022).

**Naive DocIDs.** Here, the original document identifier from a corpus is directly used and treated as a textual string. For example, a five-digit number "42915" is treated as a string and passed through the SentencePiece vocabulary of T5. Note that Naive DocIDs might sometimes capture semantics influenced by the curation pipelines that may reflect inherent relatedness.

**Semantic DocIDs.** Following Tay et al. (2022), Semantic DocIDs aim to imbue document identifiers with hierarchical semantic information instead of relying on predefined identifiers. Specifically, after encoding documents into dense vectors, a hierarchical $k$-means algorithm recursively clusters the space into $k$ clusters until individual clusters include no more than $c$ documents. Consequently, all document identifiers form a tree, where non-leaf nodes correspond to superclusters, and leaf-node clusters with at most $c$ documents each. Semantic DocIDs are formed by composing these cluster identifiers, each from 0 to $k-1$, tailed by an identifier in the leaf nodes between 0 and $c-1$. In this work, we use the identifiers generated by Wang et al. (2022) for NQ and TriviaQA to perform a fair comparison. They achieve this with a 12-layer BERT model. For MS MARCO, we use SentenceT5-Base (Ni et al., 2022a), and $c = 100$. Since the passage-level corpus is in the order of millions, if a cluster ends up with more than 1M documents, we sample 100K documents first, when computing centroids. We used $k = 10$ clusters at each level, corresponding to the ten digits $(0 \ldots 9)$.

**2D Semantic DocIDs.** In the Semantic DocID setting, the same tokens represent different semantic meanings at different positions: we use the same set of tokens between 0 to $k-1$ for all identifiers, but they represent semantic clusters at different levels in the tree. To address this, NCI (Wang et al., 2022) introduces a 2D variant by adding an extra dimension to encode the positions, making the model aware of levels of clustering when decoding the identifier. To allow better modeling with this variant, they additionally introduce a change to the decoder, Prefix-Aware Weight-Adaptive Decoding, which we describe in the next section.

## 2.3 Model Variants

Besides alternative ways of constructing model inputs and targets, generative retrieval approaches that build on DSI have also investigated novel modeling components. Here, we review three model components introduced by Wang et al. (2022) and Bevilacqua et al. (2022).

**Prefix-Aware Weight-Adaptive (PAWA) Decoder** is proposed as a method for decoding 2D Semantic DocIDs. Unlike a standard Transformer decoder, which uses the same matrix to project the hidden representation of the decoder to the vocabulary space for every position, a PAWA decoder uses different projection matrices at each timestep, with the weights of each projection matrix computed adaptively by a separate Transformer decoder. Specifically, in a vanilla decoder, the dense representation $h \in \mathbb{R}^{l \times d}$ from the last decoder layer is projected into the vocabulary space with $W \in \mathbb{R}^{d \times |V|}$, where $l$ denotes the sequence length for decoding. To incorporate the position, the extra decoding module separately processes the input query and the already-decoded DocID tokens to output a projection matrix $W^{pawa} \in \mathbb{R}^{d \times l \times |V|}$, replacing $W$. This aims to capture that the semantic meaning of a DocID token depends on its position in the output sequence as well as on the DocID prefix preceding it. The experiments in this paper use the open-source PAWA decoder implementation provided by the original authors[1] as a reference and build it out on t5x. For more details, please refer to NCI (Wang et al., 2022) and their codebase.

**Constrained decoding** can be used to avoid generating invalid document identifiers (Bevilacqua et al., 2022; Wang et al., 2022). While we have empirically found that roughly less than 1 in 20 DSI-based generation beams are invalid, we include this method nonetheless, as it is widespread in the literature. In this work, we adopt an exact match approach that leverages a trie to ensure we decode only valid document identifiers.

**Consistency loss** helps alleviate over-fitting by introducing a regularization term. The basic idea is that the representations generated by two forward passes with different dropout masks should be similar. Wang et al. (2022) incorporate

---

| Dataset | #Docs | % Covered by train query set |
|---------|-------|------------------------------|
| NQ100K (Wang et al., 2022) | 110K | 98.4% |
| TriviaQA (Wang et al., 2022) | 74K | 57.7% |
| MSMarco100K | 100K | 92.9% |
| MSMarco1M | 1M | 51.6% |
| MSMarcoFULL | 8.8M | 5.8% |

Table 1: The coverage statistics of the benchmark datasets and their training query sets.

this insight into a regularization term that augments the generation loss. We investigate the softmax version as described in the NCI paper (Eq. 5 in (Wang et al., 2022)) and a KL-divergence version from an early version of NCI (Eq. 1). They compute the Kullback-Leibler (KL) divergence between the output probabilities of two independent forward passes per position, where $p_{i,1}$ and $p_{i,2}$ are the probability distributions over the vocabulary space from the two forward passes at position $i$, respectively.

$$\mathcal{L}_{reg} = \tfrac{1}{2}[D_{KL}(p_{i,1} \| p_{i,2}) + D_{KL}(p_{i,2} \| p_{i,1})] \quad (1)$$

While we closely follow the implementation of the Neural Corpus Indexer codebase, we find that these regularization terms lead to training instability and that the model effectiveness often diverges into a $NaN$ loss. As a result, we do not include consistency regularization in our final experiments.

## 3 Experimental Setting

We limit ourselves to English retrieval tasks, focusing on the behavior of generative retrieval models at varying corpus scales.

### 3.1 Corpus and Training Data

Following small-scale generative retrieval experiment setups (Tay et al., 2022; Wang et al., 2022; Zhuang et al., 2022b; Chen et al., 2023), we start with experiments on the Natural Questions (Kwiatkowski et al., 2019) and TriviaQA (Joshi et al., 2017) datasets. To better understand how different model configurations perform at scale and in more practical settings, we also experiment with variants of the MS MARCO passage ranking dataset. The MS MARCO passage ranking collection consists of a corpus of 8.8M passages and a training set of 532K queries. From this dataset, we construct three variants, namely, MSMarco100K (100K passages), MSMarco1M (1M passages), and MSMarcoFULL (all 8.8M passages). It is worth noting that most documents in NQ100K and MSMarco100K have at least one relevant query in the training set. However, as we scale to MSMarcoFULL, the fraction of documents with queries in the training set drastically drops to around 6%, leading to a more practical setup. We summarize the statistics of these datasets in Table 1.

**NQ100K & TriviaQA.** To enable comparisons, we reuse the documents, the segmented documents, the training/testing splits, and generated query sets from Wang et al. (2022). The Natural Questions and TriviaQA datasets have corpora of sizes 109K and 74K, respectively. Note that Wang et al. (2022) refers to NQ100K as NQ320K; we refer to the number of *unique* documents instead of the labeled training data size. Most documents in the NQ100K dataset have at least one relevant question in the training data, while 58% of the TriviaQA dataset has this property.

**MSMarco100K, 1M, & FULL.** In the same vein as NQ100K and TriviaQA, for MSMarco100K, we curate a dataset with 100K passages sampled from the full MSMarco passage ranking dataset. Most passages have at least one positive query for training. We also include passages relevant to the queries in the development dataset (for evaluation). MSMarco1M is $10\times$ larger than MSMarco100K. As with MSMarco100K, we augment the corpus with passages relevant to development queries. We first include all passages relevant to the 533K and 7K queries from the training dataset and development sets, respectively. This results in 516K and 7K unique passages from each set. We randomly sample passages without a query in either set to total a million passages. Finally, MSMarcoFULL scales up another order of magnitude in corpus size. As a result, only 5.8% of the passages have a corresponding query in the training set. We aren't aware of any previous work that has attempted to apply generative retrieval models to a dataset of this size and complexity.

### 3.2 Synthetic Query Generation

For NQ100K and TriviaQA, we reuse the generated questions from (Wang et al., 2022) with 20 and 15 generated questions for each document, respectively. For the MSMarco variants, we use docT5query (Nogueira and Lin, 2019) to

generate questions, with 40 generated questions per passage. We also train a question-generation model using T5-base using training data from DPR (Karpukhin et al., 2020), a retrieval dataset derived from NQ (Kwiatkowski et al., 2019). We use this model to generate 40 questions per passage, following the configuration of docT5query. We refer to this setting for NQ and TriviaQA as "in-domain D2Q".

## 3.3 Evaluation Dataset and Metrics

We evaluate using the validation set of each dataset. For NQ100K, TriviaQA, and MSMarco, the set contains 7,830, 7,993, and 6,980 queries, respectively. We use the same evaluation set for all three MSMarco variants considered. For each query in the evaluation sets, we use the models to generate ranked lists of documents. We report Recall@1, Recall@5, and MRR@10 (Mean Reciprocal Rank) for NQ, TriviaQA, and MSMarco, respectively.

## 3.4 Model Variants

We evaluate all methods using a T5.1.1 backbone (Raffel et al., 2020). See Appendix A.2 for implementation details, including hyperparameter settings and compute cost. We test variants of labeled vs. synthetic queries, FirstP vs. DaQ document representations, and combinations of multiple representation choices. For each model variant, we ablate all versions of document identifiers when applicable. Modifications in model architecture are introduced in a stacking fashion, starting with the base model and then adding on the PAWA decoder, constrained decoding, and consistency loss in this order. Note, we only evaluate the PAWA decoder with 2D Semantic DocIDs as Wang et al. (2022) introduce the decoder to exploit their unique representation hierarchy.

For model scaling experiments, we mainly investigate whether Atomic DocIDs are an effective way to scale to millions of passages, given the parameter cost. As such, we consider larger models with Naive DocIDs and Semantic DocIDs comparable to T5-Base with Atomic DocIDs, which total 7B parameters when scaling to 8.8M DocIDs.

For baselines we provide BM25 (Robertson and Zaragoza, 2009) and BM25 with doc2query-T5 (Nogueira and Lin, 2019). For Natural Questions and TriviaQA, we also include the previous results reported for the NCI-variant of NQ (i.e., NQ100K). The baselines include state-of-the-art generative retrieval results like NCI and GenRet (Sun et al.,

2023), and GTR-Base, a state-of-the-art dual encoder (Ni et al., 2022b). For the MSMarco variants we introduce, we provide our own GTR-Base (Ni et al., 2022b) results.

## 4 Experimental Results

We report our results in three parts. First, we ablate all the methods from Section 2 using T5-Base on small-scale datasets: NQ100K and TriviaQA. We observe which techniques work best on this small scale with widely studied datasets. Then, we transfer the same techniques and scale up to the entire MS MARCO passage ranking dataset. Finally, to understand whether we can attribute the effectiveness bump from Atomic DocIDs to additional model parameters, we fix the corpus to be MSMarcoFULL and scale the model size up to 11B. Across the tables, At. refers to Atomic DocIDs, Nv. Naive DocIDs, and Sm. Semantic DocIDs.

## 4.1 Ablations over Small Corpora

We report our ablations over NQ100K and TriviaQA in Table 2. The best combination of our techniques (row (7)) sets a new state-of-the-art result on the NCI variant of Natural Questions without using sophisticated modeling techniques, architectural changes, and learned DocIDs. The choice of document representation by far dominates the overall effectiveness of the retriever. Using just the training queries provided by the datasets demonstrates poor results due to the low coverage of the documents. With FirstP, we see considerable improvements over this setting. The DaQ representation strategy further improves over FirstP, exposing the model to different views of the same document. However, synthetic queries from D2Q are essential to high generative retrieval effectiveness, resulting in a gain of more than 7 points. This technique, by far, trumps all other proposed changes.

As for other design choices, for this small-scale setup, Naive and Semantic DocIDs compete on par with each other (varying between task configurations), and Atomic DocIDs are consistently the best. We note, though, that on NQ100K, Atomic DocIDs add 80M parameters to a T5-Base model that would otherwise be 220M parameters (a 36% increase). Given the comparable effectiveness in the best configuration (row (7)), these extra parameters may or may not be worth it, but we refer to Section A.3.2 for more discussion. Modeling techniques from (Wang et al., 2022), i.e., 2D Semantic

| Model | NQ100K | | | TriviaQA | | |
|---|---|---|---|---|---|---|
| | At. | Nv. | Sm. | At. | Nv. | Sm. |
| *Baselines* | | | | | | |
| BM25 (via Wang et al. (2022)) | - | 15.1 | - | - | 56.9 | - |
| BM25 w/ doc2query–T5 (via Wang et al. (2022) | - | 35.4 | - | - | 59.7 | - |
| GTR-Base (via Sun et al. (2023)) | - | 56.0 | - | - | - | - |
| NCI (Wang et al., 2022) | - | 62.8 | 65.9 | - | 88.8 | 90.5 |
| GenRet (Sun et al., 2023) | - | - | 68.1 | - | - | - |
| *Ours* | | | | | | |
| (1a) Labeled Queries (No Indexing) | 50.7 | 49.2 | 49.0 | 60.9 | 56.7 | 61.4 |
| (2a) FirstP + Labeled Queries (DSI) | 60.0 | 58.4 | 58.7 | 71.6 | 75.2 | 78.9 |
| (2b) DaQ + Labeled Queries | 61.4 | 60.4 | 60.0 | 81.0 | 80.4 | 77.6 |
| (3a) DaQ + D2Q + Labeled Queries | 69.6 | 67.9 | 67.9 | 88.2 | 85.7 | 86.3 |
| (3b) FirstP + DaQ + D2Q + Labeled Queries | 69.0 | 68.2 | 67.2 | 88.9 | 86.9 | 87.4 |
| (4a) 3b + PAWA (w/ 2D Semantic DocIDs) | - | - | 66.3 | - | - | 86.5 |
| (4b) 3b + Constrained Decoding | - | - | 67.3 | - | - | 87.3 |
| (5) 4b + Consistency Loss (NCI) | - | - | 66.3 | - | - | 86.6 |
| (6a) DaQ Only | 17.1 | 18.4 | 15.6 | 41.0 | 31.3 | 20.6 |
| (6b) D2Q Only | 43.6 | 42.3 | 42.9 | 61.9 | 57.8 | 57.1 |
| (6c) 6b + PAWA (w/ 2D Semantic DocIDs) + Constrained Decoding | - | - | 43.1 | - | - | 57.7 |
| (7) 3b + in-domain D2Q | **70.7** | **69.7** | **69.5** | **90.0** | **88.0** | **89.2** |

Table 2: Results on small-scale NQ (Recall@1) and TriviaQA (Recall@5) datasets. We present ablation results in a stacking fashion. Rows 6 isolate the importance of D2Q. Last row revises the best method with in-domain D2Q.

DocIDs, PAWA, constrained decoding, and consistency loss, do not reliably improve the model over using synthetic queries alone.

At this corpus scale, our best result uses a mixture of FirstP, DaQ, labeled queries, and synthetic queries for training. However, notably, the *quality* of the synthetic queries are very important, with those from a generator we specifically trained for the question-answering domain significantly outperforming the query generator trained over MS-MarcoFULL.

## 4.2 Scaling Corpus Size

We now consider variants of the MS MARCO passage ranking task, scaling from 100K to 1M and 8.8M passages. Results are reported in Table 3. Perhaps the most striking observation about the transition to the MS MARCO passage ranking collection is the absolute requirement of synthetic queries for strong retrieval effectiveness. Synthetic queries result in a 2-3× improvement over the original DSI formulation alone. Using only synthetic queries as the indexing task is the most effective and straightforward training strategy for the MS MARCO collection. This result signifies a notable difference in the transition from NQ and TriviaQA to MS MARCO, where FirstP and DaQ did provide substantial value. The reason may be an artifact of the NQ and TriviaQA datasets leveraging a corpus

of Wikipedia articles: the beginning of Wikipedia documents are informative entity descriptions, and many sentences refer to the entity–which is likely the answer to a requested query.

As we scale the corpus size, DSI effectiveness drops rapidly, with the best result (D2Q only with Atomic DocIDs) falling off from 80.3 to 55.8 and finally 24.2 as we scale to the whole 8.8M passages. The Semantic DocID setting, too, drops off as we scale to the whole corpus, under-performing the Naive DocID setting. We conjecture that this may be due to the potentially increased length of Semantic DocIDs being more difficult to decode than Naive DocIDs coupled with a noisy partitioning of the semantic space (especially when using an off-the-shelf embedding model such as SentenceT5-Base). However, we observe Semantic DocIDs decoded via the PAWA decoder score higher. We hope to shed some insight into why this might be in the next section, where we examine model size. Constrained decoding only provides marginal value and generally is not worth the added complexity.

## 4.3 Scaling Model Size

How much of the Atomic DocID setting's effectiveness can be attributed to its additional model parameters? On MSMarcoFULL, decoding Atomic DocID tokens adds an overhead of 7B parameters to the otherwise 220M-parameter T5-Base model.

| Model | MSMarco100K | | | MSMarco1M | | | MSMarcoFULL | | |
|---|---|---|---|---|---|---|---|---|---|
| | At. | Nv. | Sm. | At. | Nv. | Sm. | At. | Nv. | Sm. |
| *Baselines* | | | | | | | | | |
| BM25 | - | 65.3 | - | - | 41.3 | - | - | 18.4 | - |
| BM25 (w/ doc2query–T5) | - | 80.4 | - | - | 56.6 | - | - | 27.2 | - |
| GTR-Base | - | 83.2 | - | - | 60.7 | - | - | 34.8 | - |
| *Ours* | | | | | | | | | |
| (1a) Labeled Queries (No Indexing) | 0.0 | 1.1 | 0.0 | 0.0 | 0.5 | 0.0 | 0.0 | 0.0 | 0.0 |
| (2a) FirstP/DaQ + Labeled Queries (DSI) | 0.0 | 23.9 | 19.2 | 2.1 | 12.4 | 7.4 | 0.0 | 7.5 | 3.1 |
| (3b) FirstP/DaQ + D2Q + Labeled Queries | 79.2 | 77.7 | 76.8 | 53.3 | 48.2 | 47.1 | 14.2 | **13.2** | 6.4 |
| (4a) 3b + PAWA (w/ 2D Semantic DocIDs) | - | - | 77.1 | - | - | 50.2 | - | - | 9.0 |
| (5) 4a + Consistency Loss (NCI) | - | - | 77.1 | - | - | 50.2 | - | - | 9.1 |
| (6b) D2Q only | **80.3** | **78.7** | 78.5 | **55.8** | **55.4** | 54.0 | **24.2** | **13.3** | 11.8 |
| (4a′) 6b + PAWA (w/ 2D Semantic DocIDs) | - | - | 78.2 | - | - | **54.1** | - | - | **17.3** |
| (4b′) 6b + Constrained Decoding | - | - | **78.6** | - | - | 54.0 | - | - | 12.0 |
| (5′) 6b + PAWA (w/ 2D Semantic DocIDs) + Constrained Decoding | - | - | 78.3 | - | - | **54.2** | - | - | **17.4** |

Table 3: Results on the development set of the scale variant MS MARCO V1 passage collections, reported in MRR@10. Best results per column and results within 0.1 of best are bolded. Note that FirstP here is equivalent to DaQ as MS MARCO input passages fit into the input window. For more metrics (nDCG@20, Hits@1,5,10,20, Precision@1,5,10,20), please see Section A.5 of the Appendix.

| T5 Scale | Training | Params | Inference FLOPs | MRR@10 |
|---|---|---|---|---|
| Base | D2Q Only + Atomic DocID | 7.0B | $0.9 \times 10^{12}$ | 24.2 |
| Base | D2Q Only + Naive DocID | 220M | $1.4 \times 10^{12}$ | 13.3 |
| Base | D2Q Only + PAWA (2D Sem.) | 761M | $6.8 \times 10^{12}$ | 17.3 |
| Large | D2Q Only + Naive DocID | 783M | $3.5 \times 10^{12}$ | 21.4 |
| Large | D2Q Only + PAWA (2D Sem.) | 2.1B | $1.1 \times 10^{13}$ | 19.8 |
| XL | D2Q Only + Naive DocID | 2.8B | $9.3 \times 10^{12}$ | **26.7** |
| XXL | D2Q Only + Naive DocID | 11B | $4.3 \times 10^{13}$ | 24.3 |

Table 4: The effect of scaling up model size for sequential DocID approaches in comparison to Atomic DocIDs for the MSMarcoFULL task.

We take the best configuration on MSMarcoFULL from Table 3 and scale model parameters of Naive DocID and Semantic DocID (PAWA) to similar sizes for comparison. We report results in Table 4.

Overall, we observe a general trend that as parameter count increases, retrieval effectiveness improves. Indeed, both Atomic DocIDs and PAWA Semantic DocIDs had the highest scores in Table 3, which we now attribute to their increased size. Notice that the difference here only comes out when scaling to MSMarcoFULL, where these parameter differences magnify significantly over corpora of a smaller scale. However, not all methods are equal. PAWA and 2D Semantic DocIDs (Wang et al., 2022) significantly increase decoding parameters with its extra decoding stack, yet yield no gain over naively scaling the Transformer with Naive DocIDs, underperforming by 4 points at around ∼800M parameters. This pattern continues to hold scaling PAWA to 2.1B parameters. Thus, to conserve resources, we do not scale PAWA any further.

Scaling Transformers naively according to default T5 scales while using Naive DocIDs had the highest effectiveness on MSMarcoFULL at 26.7 MRR@10. Despite using only ∼2.8B parameters, this approach outperforms T5-Base with Atomic DocIDs, which uses 7B parameters while achieving only 24.2 MRR@10. However, while parameter count has practical implications regarding the resources required for training and inference (especially TPU/GPU memory), there are other trade-offs to consider, which we discuss in the Appendix A.3.2.

While Naive DocIDs perform well at T5-XL size, surprisingly, we find that scaling further to XXL (11B) does not improve effectiveness; in fact, it is detrimental to retrieval effectiveness (24.3 MRR@10 vs. XL's 26.7) under the same experimental settings and hyperparameter settings, even though model training converges faster. This finding is counter-intuitive to most generation tasks and the common conception of generative retrieval re-

| Model / T5 Scale | Training | Params | MSMarcoFULL Recall@100 | MSMarcoFULL nDCG@10 | TREC DL 19 nDCG@10 | TREC DL 20 nDCG@10 |
|---|---|---|---|---|---|---|
| *Baselines* | | | | | | |
| BM25 | – | – | 65.8 | 22.8 | – | – |
| BM25 w/ docT5query | – | – | 81.9 | 33.8 | – | – |
| GTR-Base (Ni et al., 2022b) | – | 110M | 89.8 | 42.0 | – | – |
| *Ours (T5 Scale)* | | | | | | |
| Base | D2Q Only + Atomic DocID | 7.0B | 80.5 | 29.7 | 49.8 | **53.8** |
| Large | D2Q Only + Naive DocID | 783M | 82.0 | 27.3 | 49.9 | 45.2 |
| Large | D2Q Only + PAWA (2D Sem.) | 2.1B | 73.2 | 24.8 | 46.6 | 50.1 |
| XL | D2Q Only + Naive DocID | 2.8B | **84.7** | **33.2** | **55.0** | 52.2 |
| XXL | D2Q Only + Naive DocID | 11B | 81.9 | 30.6 | 52.0 | 49.0 |

Table 5: Additional evaluation of the top five model checkpoints from Table 4 on TREC DL 19 & 20. We decode using 100 beams here. The same metrics are reported over the original MSMarco validation set, with the addition of Recall@100.

lying on model capacity for retrieval effectiveness.

To strengthen the evidence from our scaling results, we provide results on TREC DL 19 and 20 (Craswell et al., 2020, 2021) in Table 5. These additional densely judged evaluation sets, as opposed to the sparse judgments in the MS MARCO v1 Passage Ranking validation set, seek to address some of the issues with the original task. We see that our original findings generally hold under these new metrics and believe that this addition reaffirms the soundness of our study.

## 5 Analysis & Discussion

The results of this work raise major questions, including: *Why are synthetic queries effective?* and *Which model scaling approach is the best?* We present an extended analysis, including additional experiments, to provide more insight on these topics in Section A.3 of the Appendix. Finally, we discuss the limitations of our study and future directions for the paradigm in Section 7 and A.4, respectively.

## 6 Conclusion

We provide the first empirical study of generative retrieval methods over the large-scale 8.8M passage corpus of the MS MARCO passage ranking task. Of the various methods from the literature that we consider in this work (Tay et al., 2022; Zhuang et al., 2022b; Wang et al., 2022), we find that the use of synthetic queries as a document representation strategy is the only approach that remained effective and necessary, as we scaled up the corpus size. We highlight the importance of accounting for the compute cost of techniques; keeping the parameter count fixed, we find that naive methods outperform more sophisticated ones on the full MS

MARCO dataset. Our best result on MS MARCO passage ranking uses only synthetic queries and Naive DocIDs in the target space for the training task, with the model scaled to T5-XL (3B). This model only achieves an MRR@10 of 26.7. Surprisingly, increasing parameters for the same setting up to XXL (11B) scores worse.

Altogether, our results highlight the unique challenges currently facing the generative retrieval paradigm, the need for closer attention to method comparisons, and more fundamental improvements before we can fully leverage the power of larger language models in the future.

## 7 Limitations

As with all empirical studies, ours has its own set of limitations that we urge the reader to consider. Multiple works have come after the experiments in this work, e.g., Chen et al. (2023), and thus, we do not present an exhaustive set of generative retrieval techniques here. For instance, the vast space of identifiers could be in natural language or learned codes. In addition, due to resource constraints, our model scaling experiments are not exhaustive, and not all ablation scenarios in Table 3 are scaled to larger model sizes. It could be possible that some missed setups improve more at larger parameterizations, albeit unlikely, as with scaling past 11B. In addition, we do not saturate the scaling of Atomic DocIDs because of the extreme parameter requirements. Finally, since this work focused on the effectiveness of generative retrieval on large corpora, scaling model size for smaller corpora was outside our scope. Investigating the maximum corpus size for which generative retrieval could provide state-of-the-art effectiveness is a question of practical importance, which we leave for future work.

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

# A Appendix

Due to space constraints, we include related work, additional discussion, analysis, experiments, and implementation details here.

## A.1 Related Work

Traditional retrieval models like BM25 (Robertson and Zaragoza, 2009) that rely on the lexical overlap, term frequency heuristics, and inverse document frequency, while reasonably strong on their own, tend to fail at matching documents that have minor word overlap but are semantically related.

A popular solution is dual encoders (Gillick et al., 2018; Karpukhin et al., 2020; Chen et al., 2022), where a pretrained language model such as BERT (Devlin et al., 2019) computes low-dimensional dense representations instead of the high-dimensional sparse representations found in BM25. Fine-tuning pretrained models on the target task can further improve the effectiveness of dual encoders. Based on the success of T5 in various natural language understanding tasks, Ni et al. (2022a) proposes scaling up dual encoders by training T5-style pretrained language models with a two-stage contrastive learning approach on the Semantic Text Similarity (STS) tasks. The Generalizable T5 Retriever (GTR) (Ni et al., 2022b) extends this idea to information retrieval. The most successful GTR models were pretrained on a large-scale question-answering dataset curated from the internet and fine-tuned on the MS MARCO Passage Ranking task (Nguyen et al., 2016).

Existing approaches often apply synthetic query generation to improve retrieval effectiveness. Nogueira et al. (2019) first leveraged a vanilla sequence-to-sequence Transformer to train a model that can map passages to queries that it might be able to answer. Nogueira and Lin (2019), doc2query-T5 further improved the effectiveness of the traditional Transformer by leveraging a T5 model. Ma et al. (2022) experimented with similar ideas and showed that query generation is effective across a wide range of corpora and task setups.

Prior to generative retrieval, sequence-to-sequence language models, like T5 (Raffel et al., 2020), were shown to be effective for reranking tasks. In this setup, models assign scores to the top-$k$ results from a first-stage retrieval method. One can then use these scores to rerank the documents. For example, monoT5 (Nogueira et al., 2020) was the first to leverage T5 as a pointwise reranker by training a model that takes the concatenation of the query and document as input and generates a relevance label. Pradeep et al. (2021b); Zhuang et al. (2022a); Hui et al. (2022) have since improved the effectiveness and efficiency of generation-based reranking. These approaches continue to demonstrate strong effectiveness in various retrieval tasks (Pradeep et al., 2021a; Craswell et al., 2022; Pradeep et al., 2022).

Generative retrieval seeks to replace the entire information retrieval process with a single sequence-to-sequence model capable of mapping queries directly to relevant document identifiers (Metzler et al., 2021). Differentiable Search Indexes (DSI) (Tay et al., 2022) first demonstrated the potential of this paradigm, where T5 is used to parameterize an end-to-end search system, with the model parameters encoding all information about the corpus. See Section 2 for more information. DSI was shown to outperform a dual encoder baseline on the Natural Questions dataset (Kwiatkowski et al., 2019). Zhuang et al. (2022b) explores the effectiveness of DSI and synthetic queries on a 100K passage subset of the MS MARCO passage ranking corpus and XOR QA (Asai et al., 2021). Neural Corpus Indexer (Wang et al., 2022) builds on the success of DSI and introduces a combination of more input variants and architectural additions, some of which we describe and explore in this work. Many works have explored various document identifier designs, including document substring (Bevilacqua et al., 2022), metadata-based approaches (Zhou et al., 2022; Ziems et al., 2023), and learned quantization (Rajput et al., 2023; Sun et al., 2023). More recently, (Chen et al., 2023) proposes a distillation approach on top of DSI, learning from the rankings generated by dense retrieval using a multi-task training loss. Note, this is not to be confused with "generative retrieval models" as used by Lesota et al. (2021), defined via the cumulative probabilities of generating query terms to provide a probabilistic perspective on relevance estimation and to offer a nuanced measure of uncertainty.

However, none of these works have explored training or evaluating generative retrieval systems on corpora larger than O(100K) documents. Given that the generative retrieval paradigm has extended beyond traditional retrieval into areas such as recommender systems (Rajput et al., 2023) and vision (Zhang et al., 2023), we believe our study will be crucial for an evergrowing community.

## A.2 Implementation Details

We use T5.1.1 as implemented by t5x (Roberts et al., 2022). We implement the different setups described in Section 2 in the form of seqio tasks. For the MS MARCO variants, we set the maximum input sequence length to 128 for all experiments. Following the NCI setup, we set it to 64 for the NQ100K and TriviaQA experiments. We initialize our models with the pre-trained T5-base model. For the PAWA decoder, we randomly initialize the PAWA model parameters. Following (Tay et al., 2022) for sequential DocIDs, beam search, with 40 beams, is used during inference.

We revise hyperparameter settings from (Tay et al., 2022) to ones we have found to empirically perform better, especially for indexing larger corpora like MSMarcoFULL. We set the batch size in all our experiments to 512. We train our models with a learning rate of $10^{-3}$ and a dropout rate of 0.1. We use 10K learning rate warm-up steps for all runs, except for Atomic DocIDs that use 100K steps. We train our small-scale datasets, NQ100K, TriviaQA, and MSMarco100K, for 1M steps. For MSMarco1M and MSMarcoFULL, we train our model to convergence or, at most, 9M steps. We use 8 TPUv4 chips for training models at the T5-Base scale. T5-Large, T5-XL, and T5-Base with Atomic DocIDs over MSMarcoFULL use 64 TPUv4 chips. For T5-XXL, we use 128 chips. Our most expensive runs took roughly 10-14 days to train to convergence on MSMarcoFULL.

## A.3 Discussion

The results of this work raise multiple questions regarding the current state of generative retrieval at a scale that we aim to provide more insight into here.

### A.3.1 Why are synthetic queries effective?

Although the use of synthetic queries as a document representation technique is effective in previous works (Zhuang et al., 2022b; Wang et al., 2022; Chen et al., 2023), our experiments highlight its central importance to generative retrieval on a larger, more challenging corpus. We suggest that the effectiveness of synthetic queries mainly comes from augmenting the input distribution during training to be closer to that observed at inference/evaluation time. Mainly, this comes in two forms: mitigating the coverage gap of ground-truth labeled queries and the document corpus and closing the gap between the training query distribution

and inference/evaluation. In addition, we find that the diversity of generated synthetic queries can impact retrieval effectiveness.

**Document coverage gap.** In Table 1, for each dataset considered, we report the coverage of their document corpus by the corresponding labeled query training set. When comparing MSMarco100k, 1M, and FULL, the query coverage drops from a 92.9% to a 51.6% and a 5.8%, respectively. Consider rows (2a) and (3b) in Table 3, which only differ by the addition of synthetic queries. In these settings, we observe that MSMarco100K improved by $3.3\times$ while MSMarco1M improved by 3.9x, even though 1M is a larger corpus and may be affected by model capacity, as we see with MSMarcoFULL. Similarly, for NQ100k and TriviaQA, which have 98.4% and 57.7% coverage, respectively, we observe that swapping Labeled Queries (No Indexing) (row 1a) for D2Q only (row 6b) hurts performance for NQ100k while improving performance for TriviaQA (Table 2). Since the D2Q model is trained on MS MARCO, for NQ100K, replacing its labeled training queries with synthetic queries only amounted to a 1.6% coverage gain, which is not worth the domain shift. However, for TriviaQA, this amounted to a 42.3% coverage gain, which is more worth the domain shift.

**Query distribution gap.** Synthetic query generation effectively closes the query distribution gap between training and evaluation. In Table 2, row (7) first shows the importance of the query distribution by using an in-domain query generation model to improve retrieval performance. To further understand the relationship between retrieval performance and query distribution gap, we plot the relationship between synthetic query similarity vs. validation query similarity and retrieval performance (MRR@10). For each evaluation query in the MS MARCO validation set, we measure the maximum similarity among all synthetic queries generated for the corresponding passage. We used Jaccard similarity for its simplicity. For each evaluation query, we evaluate MRR@10 using the Atomic DocID variant of row 6b in Table 3. Figure 1 reports the average MRR@10 within each 10pt Jaccard similarity bucket. We plot two variants using 40 and 100 sampled queries per passage for comparison.

In general, higher Jaccard similarity correlates with higher MRR@10 scores. That is, the more similar our training queries are to the evaluation

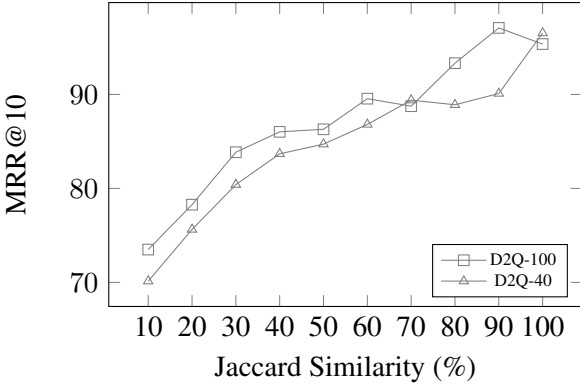

Figure 1: Jaccard similarity between synthetic queries and validation set queries vs. MRR@10 on the MS-Marco100K subset.

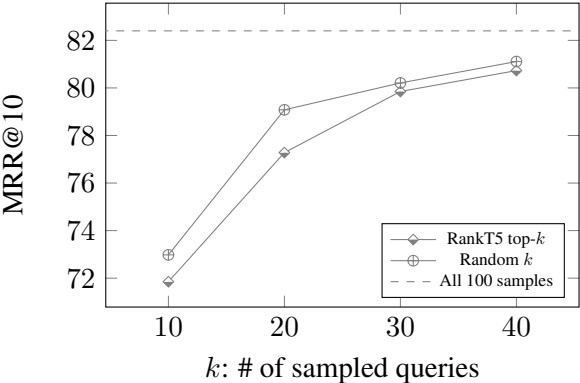

Figure 2: For MSMarco100K, we present the MRR@10 scores as we vary the number of synthetic queries per passage. Given 100 pre-generated queries per passage, we compare random-$k$ sampling, top-$k$ selection via RankT5-XL, and using all 100 synthetic queries.

ones, the better the retrieval effectiveness. Comparing the two settings, higher exposure to synthetic queries typically results in better scores across the similarity buckets. Even though the query distribution is vital, it is worth noting that even on the lowest end of similarity, this setting still has strong retrieval effectiveness. While synthetic query distribution is a crucial aspect of retrieval effectiveness, it is not singular in determining the end effectiveness, and the generative retrieval model goes far beyond simply detecting lexically similar queries to those seen during training.

**Diversity.** We provide further analysis regarding the importance of synthetic query diversity. Here, we assume the same MSMarco100K setting using the Atomic DocID variant of row 6b in Table 3. We vary the number of sampled synthetic queries per passage used for training and observe MRR@10.

We consider using 10, 20, 30, 40, and 100 sampled queries per passage, which we construct by first sampling the full 100 queries and then taking random subsets of varying sizes. We use a sampling temperature of 1.0 and consider the top 10 tokens at each sampling step. Recent studies show advances in utilizing cross encoders to refine the generated query set of incoherent, unspecific queries to improve the use of D2Q (Gospodinov et al., 2023). Accordingly, we also experiment with ranking the 100 sampled queries and taking top-10,20,30,40 instead of randomly sampling. We achieve this using a state-of-the-art cross-attention re-ranker, RankT5-XL (Zhuang et al., 2022a), to score (generated query, passage) pairs and then take the top-k.

We report results in Figure 2. We consistently find that sampling more synthetic queries improves effectiveness in this setting. Surprisingly, applying RankT5-based selection over the samples hurt scores. This suggests an overall preference for more samples and more diverse samples to improve effectiveness. Using all 100 samples scored the best, increasing MRR@10 from 80.3 (Table 3, which used 40 samples) to 82.4, closing the gap with GTR-Base (83.2 MRR@10) on MS-Marco100K. Exactly why query diversity is so important is still up for interpretation, but there could be a couple of possibilities: more diverse samples give a higher chance of at least some being close to the target distribution, and more examples could provide a type of regularization to the model.

### A.3.2 Which model scaling approach is best?

Much of this paper has considered parameter cost as a proxy for memorization capacity, which has been conjectured, in the past, to be important for retrieval (Tay et al., 2022). However, model comparisons should not stop at parameter counts as this may not correlate with other cost indicators (training speed, FLOPs, etc.) that are important to practical applications (Dehghani et al., 2022). While ultimately, the best method to scale generative retrieval models will be the one that unlocks the potential of the paradigm to be competitive on large-scale retrieval tasks, we can provide some first glimpses into what trade-offs are at stake as we consider larger models for larger corpora.

As a case study, we consider T5-Base with Atomic DocIDs compared to T5-XL with Naive DocIDs from Table 4. Both are trained only with synthetic queries and represent the only two viable approaches based on our experiments. The PAWA

decoder severely underperforms quality-wise as we scale model size, not to mention the FLOP expense of having an extra decoding stack during inference. We further discuss parameter cost, training speed, and inference FLOPs here.

*Parameters.* As corpus size scales, generative retrieval models face a fundamental prerequisite in model size to achieve decent performance, as seen in Table 3. Between the three different ways of adding parameters (naive scaling, Atomic DocIDs, PAWA decoder), we see quality improvements over the smaller models. As discussed, on a fixed parameter budget basis, Naive DocIDs demonstrate the highest effectiveness on MSMarcoFULL.

*Training Speed.* Applications that require frequent retraining value fast total training wall time. We train T5-Base Atomic DocIDs and T5-XL Naive DocIDs on the same hardware (64 TPUv4) and hyperparameter settings. To achieve the optimal performance reported in Table 4, T5-XL Naive DocIDs required 14 days while T5-Base Atomic DocID required only ∼7 days. However, at ∼7 days T5-XL Naive DocIDs were quality matched with T5-Base Atomic DocIDs (∼ 24.5 MRR@10), making both approaches roughly equal in terms of training wall-time when accounting for quality.

*Inference FLOPs.* Inference FLOPs can be a proxy for serving performance, although imperfect. Here, we see that while Sequential DocIDs can achieve more with fewer parameters, Atomic DocIDs are incredibly FLOP efficient during inference. T5-Base with Atomic DocIDs for MSMarcoFULL requires only 9.7% the inference FLOPs of T5-XL with Naive DocIDs for 90% of the retrieval performance (Table 4). How is this possible? Atomic DocIDs incur additional compute costs to calculate an output projection and softmax over the enormous vocab of ∼8.8M DocIDs. However, it only has to compute this once to get a complete ranking of the *entire* corpus — a potentially unique property of the approach. On the other hand, Sequential DocIDs require $d$ decoding steps to decode a single DocID and $k$ beams to find a ranking of $k$ DocIDs ($k = 40$ for our experiments). Thus, even though Atomic DocIDs require an expensive output projection, Sequential DocIDs require $O(d \cdot k)$ more decoding steps. To scale Naive DocIDs to be competitive with Atomic DocIDs, also makes each decoding turn significantly more expensive.

Finally, we cannot say which approach is the best as the paradigm has yet to achieve competitive results on the MS Marco passage ranking. On small corpora (100K passages), Atomic DocIDs are the highest quality, efficient option without incurring too many extra parameters. Though we can see that training models to maximize memorization amplify compute trade-offs from our experiments, the field must provide more nuanced discussions of cost trade-offs as it considers more realistic applications of generative retrieval.

## A.4 Future Directions

While open problems in generative retrieval have not changed (e.g., how to achieve state-of-the-art results on large corpora, how to update such a model with new documents (Mehta et al., 2022)), we believe that our work also raises new open questions for the field. (1) How do we properly leverage large language models and the power of scaling model parameters to benefit generative retrieval on large corpora? While Tay et al. (2022) showed this possibility over NQ, the same is not yet observed on MS MARCO even though intuitively expanded model capacity *should* benefit increased corpus scale. (2) How can we design model scaling recipes and derive scaling laws that maximize retrieval performance? In this work, we only consider default T5 parameterizations, which may or may not be optimal for memorization-heavy tasks. (3) How can we design architectures that can interpolate between the compute trade-offs of Atomic DocIDs and Sequential DocIDs? We look forward to understanding more about these vital problems to the success of generative retrieval in future work.

## A.5 Additional Results

In the remaining pages, we include extended results. Besides MRR@10 in Table 3, we report nDCG@20, Hits@K, and Precision@K in Table 6 for our experiments on MSMarco, corresponding to different rows in Table 3 indexing with the row id. We can see that these metrics correlate well with MRR@10, and our observations remain consistent with those already described.

| Rows in Table 3 | Dataset | DocID | nDCG@20 | Hits@1 | Hits@5 | Hits@10 | Hits@20 | P@1 | P@5 | P@10 | P@20 |
|---|---|---|---|---|---|---|---|---|---|---|---|
| 2a | MSMarco100K | Atomic | 0.0 | 0.01 | 0.01 | 0.03 | 0.03 | 0.01 | 0.00 | 0.00 | 0.00 |
| 2a | MSMarco100K | Naive | 26.9 | 19.23 | 30.10 | 33.95 | 38.21 | 19.23 | 6.17 | 3.49 | 1.97 |
| 2a | MSMarco100K | Semantic | 24.7 | 15.52 | 27.58 | 33.48 | 39.00 | 15.52 | 5.57 | 3.40 | 1.99 |
| 2a | MSMarco1M | Atomic | 4.1 | 1.33 | 4.20 | 6.17 | 9.17 | 1.33 | 0.84 | 0.62 | 0.46 |
| 2a | MSMarco1M | Naive | 14.7 | 9.13 | 16.83 | 19.84 | 23.12 | 9.13 | 3.44 | 2.03 | 1.19 |
| 2a | MSMarco1M | Semantic | 11.2 | 3.91 | 12.16 | 17.77 | 24.17 | 3.91 | 2.46 | 1.80 | 1.23 |
| 2a | MSMarcoFULL | Atomic | 0.3 | 0.19 | 0.27 | 0.33 | 0.42 | 0.19 | 0.05 | 0.03 | 0.02 |
| 2a | MSMarcoFULL | Naive | 13.3 | 3.31 | 13.75 | 23.44 | 31.13 | 3.31 | 2.83 | 2.43 | 1.62 |
| 2a | MSMarcoFULL | Semantic | 5.5 | 1.32 | 5.40 | 9.01 | 13.28 | 1.32 | 1.09 | 0.91 | 0.67 |
| 3b | MSMarco100K | Atomic | 83.0 | 71.42 | 90.21 | 93.34 | 95.24 | 71.42 | 18.86 | 9.83 | 5.03 |
| 3b | MSMarco100K | Naive | 82.2 | 72.08 | 88.11 | 91.45 | 93.45 | 72.08 | 18.52 | 9.65 | 4.94 |
| 3b | MSMarco100K | Semantic | 82.0 | 71.26 | 88.44 | 91.58 | 93.81 | 71.26 | 18.49 | 9.63 | 4.95 |
| 3b | MSMarco1M | Atomic | 62.0 | 43.78 | 70.82 | 78.84 | 85.14 | 43.78 | 14.61 | 8.18 | 4.45 |
| 3b | MSMarco1M | Naive | 59.5 | 42.61 | 67.92 | 75.09 | 81.02 | 42.61 | 14.09 | 7.83 | 4.24 |
| 3b | MSMarco1M | Semantic | 59.0 | 41.65 | 66.73 | 74.73 | 81.50 | 41.65 | 13.78 | 7.76 | 4.26 |
| 3b | MSMarcoFULL | Atomic | 31.1 | 13.54 | 36.28 | 47.78 | 58.12 | 13.54 | 7.35 | 4.87 | 2.99 |
| 3b | MSMarcoFULL | Naive | 23.8 | 5.90 | 25.42 | 42.31 | 54.40 | 5.90 | 5.19 | 4.39 | 2.83 |
| 3b | MSMarcoFULL | Semantic | 10.8 | 3.12 | 11.42 | 17.32 | 24.58 | 3.12 | 2.30 | 1.74 | 1.24 |
| 4a | MSMarco100K | Semantic | 80.3 | 69.57 | 86.63 | 90.44 | 92.62 | 69.57 | 18.07 | 9.47 | 4.87 |
| 4a | MSMarco1M | Semantic | 56.9 | 38.97 | 64.91 | 73.62 | 80.27 | 38.97 | 13.39 | 7.63 | 4.18 |
| 4a | MSMarcoFULL | Semantic | 13.3 | 4.80 | 14.71 | 20.56 | 28.02 | 4.80 | 2.97 | 2.08 | 1.42 |
| 5 | MSMarco100K | Semantic | 80.4 | 69.66 | 86.82 | 90.59 | 92.74 | 69.66 | 18.11 | 9.49 | 4.88 |
| 5 | MSMarco1M | Semantic | 57.0 | 38.98 | 64.87 | 73.84 | 80.47 | 38.98 | 13.38 | 7.64 | 4.19 |
| 5 | MSMarcoFULL | Semantic | 13.5 | 4.84 | 14.80 | 20.83 | 28.62 | 4.84 | 2.98 | 2.11 | 1.45 |
| 6b | MSMarco100K | Atomic | 83.6 | 72.85 | 90.33 | 93.55 | 95.44 | 72.85 | 18.91 | 9.84 | 5.05 |
| 6b | MSMarco100K | Naive | 81.9 | 71.65 | 88.24 | 91.20 | 93.04 | 71.65 | 18.53 | 9.62 | 4.92 |
| 6b | MSMarco100K | Semantic | 81.8 | 70.96 | 88.35 | 91.69 | 93.91 | 70.96 | 18.50 | 9.66 | 4.96 |
| 6b | MSMarco1M | Atomic | 62.8 | 44.11 | 71.56 | 79.87 | 86.36 | 44.11 | 14.82 | 8.31 | 4.52 |
| 6b | MSMarco1M | Naive | 61.7 | 44.76 | 69.58 | 77.12 | 82.81 | 44.76 | 14.52 | 8.07 | 4.35 |
| 6b | MSMarco1M | Semantic | 60.6 | 42.99 | 68.81 | 76.83 | 82.75 | 42.99 | 14.23 | 8.00 | 4.33 |
| 6b | MSMarcoFULL | Atomic | 32.5 | 14.27 | 37.26 | 49.64 | 60.43 | 14.27 | 7.56 | 5.07 | 3.11 |
| 6b | MSMarcoFULL | Naive | 21.9 | 5.82 | 22.42 | 37.28 | 50.16 | 5.82 | 4.59 | 3.86 | 2.62 |
| 6b | MSMarcoFULL | Semantic | 17.5 | 6.23 | 19.01 | 27.35 | 36.99 | 6.23 | 3.83 | 2.77 | 1.88 |
| 4a′ | MSMarco100K | Semantic | 81.7 | 70.72 | 88.12 | 91.56 | 94.20 | 70.72 | 18.42 | 9.62 | 4.97 |
| 4a′ | MSMarco1M | Semantic | 60.9 | 43.61 | 68.83 | 77.23 | 83.54 | 43.61 | 14.23 | 8.02 | 4.36 |
| 4a′ | MSMarcoFULL | Semantic | 24.4 | 9.37 | 27.74 | 38.28 | 48.08 | 9.37 | 5.61 | 3.89 | 2.45 |
| 4b′ | MSMarco100K | Semantic | 81.7 | 70.97 | 88.11 | 91.40 | 93.71 | 70.97 | 18.44 | 9.62 | 4.95 |
| 4b′ | MSMarco1M | Semantic | 60.6 | 42.95 | 68.74 | 76.72 | 82.91 | 42.95 | 14.21 | 7.98 | 4.34 |
| 4b′ | MSMarcoFULL | Semantic | 17.9 | 6.36 | 19.41 | 27.87 | 37.66 | 6.36 | 3.91 | 2.82 | 1.91 |
| 5′ | MSMarco100K | Semantic | 24.8 | 6.56 | 15.60 | 35.53 | 66.66 | 6.56 | 3.19 | 3.65 | 3.46 |
| 5′ | MSMarco1M | Semantic | 13.1 | 9.44 | 14.26 | 16.32 | 18.81 | 9.44 | 2.86 | 1.64 | 0.95 |
| 5′ | MSMarcoFULL | Semantic | 24.5 | 9.50 | 27.49 | 38.25 | 48.22 | 9.50 | 5.57 | 3.90 | 2.46 |

Table 6: Results with additional metrics for our experiments on variants of the MS MARCO passage ranking collection considered in Table 3.