# OpenReview forum: "How Does Generative Retrieval Scale to Millions of Passages?"
_EMNLP/2023/Conference — EMNLP 2023 Main_

### Official Review · Reviewer_JoZg · 2023-08-06

**Soundness:** 3

**Excitement:**

3: Ambivalent: It has merits (e.g., it reports state-of-the-art results, the idea is nice), but there are key weaknesses (e.g., it describes incremental work), and it can significantly benefit from another round of revision. However, I won't object to accepting it if my co-reviewers champion it.

**Missing References:**

A Modern Perspective on Query Likelihood with Deep Generative Retrieval Models
Lesota O., Rekabsaz N., Cohen D., Grasserbauer K., Eickhoff C., Schedl M.
In proceedings of the 7th ACM SIGIR International Conference on the Theory of Information Retrieval (ICTIR), July 2021

**Paper Topic And Main Contributions:**

Te work conducts an empirical study on generative retrieval - a timely topic with growing interests in the community - with the particular focus on examining the existing methods in a large scale setting. The paper reviews and categorizes various existing methods based on the ways of document representation, synthetic query generation, and creating document identifiers. Moving beyond middle-scale Q/A datasets, the study evaluates the methods on MS MARCO Passage retrieval, leveraging the T5 models with various sizes. The findings highlights the effectiveness of query generation, and suggest possible future directions for this topic.

**Questions For The Authors:**

What different do you think the findings would be, when the methods are evaluated on other large-scale datasets?

**Reasons To Accept:**

- Paper is well-written and decently covers various generative ranking models. As an empirical study, it well positions the state of the research in this field, and also suggests possible ways to move forward.
- The paper studies the large corpus setting, largely ignored in previous studies.
- The findings are interesting and could be valuable for the researchers in the field.

**Reasons To Reject:**

My main concern is that (in terms of large-scale IR collections) the study limits itself to MS MARCO Passage retrieval. As mentioned by the study and also in several previous works, the dataset has its own limitations, and the achieved results might not draw a full picture but reflect the specificities of this dataset. As an empirical study, it would be great to bring in other large dataset with different characteristics (such as TripClick) to be able to reach to stronger conclusions.

**Reproducibility:**

4: Could mostly reproduce the results, but there may be some variation because of sample variance or minor variations in their interpretation of the protocol or method.

**Reviewer Confidence:**

3: Pretty sure, but there's a chance I missed something. Although I have a good feel for this area in general, I did not carefully check the paper's details, e.g., the math, experimental design, or novelty.

---

> ### Author Rebuttal · Authors · 2023-08-28
>
> Thank you for your time in reading and reviewing our paper! We’re glad you recognize the contributions we put forth in this paper. We believe that this is an especially important contribution to the field given the relatively large number of newly proposed methods that might overstate the effectiveness of generative retrieval approaches by omitting larger corpora often found in the information retrieval community.
>
> Re: Additional datasets. Although given the time required to train the models (see Section A.3.2, i.e. 7-14 days) it is not possible to index additional datasets during the response period, we did want to present some more metrics and evaluations on other datasets to provide a more comprehensive overview.
>
> The following table expands on the top-scoring methods in our paper from Table 4 (MSMarcoFULL), and also adds two additional sets widely considered in the IR community (TREC DL19/20):
>
> | Model                              | Params | MRR@10 | Recall@100 | NDCG@10 | NDCG@100 | TREC DL 19 NDCG@10 | TREC DL 19 NDCG@100 | TREC DL 20 NDCG@10 | TREC DL 20 NDCG@100 |
> |------------------------------------|--------|--------|------------|---------|----------|--------------------|---------------------|--------------------|---------------------|
> | Base D2Q Only + Atomic ID          | 7.0B   |   24.2       | 80.47   | 29.73    | 36.35              | 49.75               | 50.21              | 53.75               | 56.01 |
> | Large D2Q Only + Naive ID          | 783M   |   21.4       | 82.03   | 27.26    | 34.32              | 49.92               | 46.1               | 45.16               | 49.18 |
> | Large D2Q Only + PAWA (2D Sem.)    | 2.1B   |     19.8       | 73.21   | 24.79    | 31.3               | 46.57               | 45.39              | 50.13               | 50.56 |
> | XL D2Q Only + Naive ID             | 2.8B   |    26.7       | 84.74   | 33.17    | 39.22              | 54.95               | 51.63              | 52.19               | 54.07 |
> | XXL D2Q Only + Naive ID            | 11B    |   24.3       | 81.87   | 30.56    | 36.96              | 52.05               | 47.67              | 49.03               | 52.1 |
>
>
>
>
> These additional densely judged evaluation datasets, as opposed to the sparse judgments in the MS MARCO v1 Passage Ranking Task, seek to address some of the issues with the original task. We see that our original findings generally hold under these new metrics, and believe that this addition reaffirms the soundness of our study.
>
> For these top Table 4 runs we additionally have {Recall, NDCG, Precision}@{1, 5, 10, 20, 50, 100} and MRR@100. Additionally, for runs in Table 3, we also have Recall@{1, 5, 10, 20}, NDCG@10, and Precision@{1, 5, 10, 20} values. We do not include them here due to the sheer size of the tables, but we plan to include them in the Appendix for our Camera Ready version. If there is a particular comparison you’re interested in on a certain metric we’d be happy to accommodate it during the discussion period. We hope that this further reinforces our findings for the community, given we are the first to tackle this task at this scale with extensive experimentation.
>
> As a final note, we do want to add that regardless of the flaws with MS MARCO, it is nevertheless important for retrieval models to work on a variety of possible use cases and be robust to various datasets. Our results show that generative retrieval systems still have a ways to go to perform well on the class of queries present in MS MARCO, a dataset that has been widely studied by the IR field.
>
> Again, thank you for your comments! These changes will surely strengthen our paper.

---

### Official Review · Reviewer_Gmvd · 2023-08-11

**Soundness:** 3

**Excitement:**

3: Ambivalent: It has merits (e.g., it reports state-of-the-art results, the idea is nice), but there are key weaknesses (e.g., it describes incremental work), and it can significantly benefit from another round of revision. However, I won't object to accepting it if my co-reviewers champion it.

**Missing References:**

Some studies that might be considered for discussion:

Li, Yongqi, Nan Yang, Liang Wang, Furu Wei, and Wenjie Li. “Multiview Identifiers Enhanced Generative Retrieval.” In Proceedings of the 61th Annual Meeting of the Association for Computational Linguistics. Association for Computational Linguistics, 2023.

Tang, Yubao, Ruqing Zhang, Jiafeng Guo, Jiangui Chen, Zuowei Zhu, Shuaiqiang Wang, Dawei Yin, and Xueqi Cheng. “Semantic-Enhanced Differentiable Search Index Inspired by Learning Strategies.” In Proceedings of the 29th ACM SIGKDD Conference on Knowledge Discovery & Data Mining. ACM, 2023.

**Paper Topic And Main Contributions:**

In this paper, the authors conduct an empirical study of generative retrieval methods, providing a detailed comparison and analysis of the impacts of modules on corpora of different scales. The authors consequently uncover several findings about generative retrieval methods.

**Reasons To Accept:**

1. The authors conduct a detailed empirical study of generative retrieval methods about corpora scales, model scales, modules, etc.
2. The authors provides some interesting insights and also point out several valuable research directions, which makes a significant contribution to the field.

**Reasons To Reject:**

1. The paper lacks of novelty in terms of methodology. After such a detailed analysis, a better method based on the findings is encouraged.
2. Some statements in this paper lack detail and analysis.
   - In Line 123, "Methods that **implicitly** increase model parameters perform better using the same T5 initialization.". What does the "implicitly" refer to, and what will the corresponding "explicitly" refer to?
   - In Line 140, "We find this counter-intuitive to the common conception of generative retrieval being limited by model **capacity**". What does the model capability here refer to, and why is the capability of a larger model (11B parameters) low?
3. The experiments are not very convincing.
   - Only one metric is compared on each dataset. Additionally, on large-scale corpora, the "@10" metric may be overly strict, and the author should consider adding metrics based on "@100".
   - Some related work (i.e., substring-based docids) has been omitted, which is also a major research branch in the field of generative retrieval, e.g., SEAL (Bevilacqua et al., 2022). I think a comparative analysis and discussion should also be carried out.
4. In Tables, some abbreviations are used without explanation of their meanings, e.g., At., D2Q. A clear explanation is suggested to provide in summary.

**Reproducibility:**

4: Could mostly reproduce the results, but there may be some variation because of sample variance or minor variations in their interpretation of the protocol or method.

**Reviewer Confidence:**

5: Positive that my evaluation is correct. I read the paper very carefully and I am very familiar with related work.

---

> ### Author Rebuttal · Authors · 2023-08-28
>
> We appreciate your thorough review and constructive feedback. We would like to address your concerns as follows:
>
> 1. Novelty and Methodology: While we understand your concern about the lack of a new method, the primary aim of our paper is to conduct an empirical study of generative retrieval methods at scale, which is currently poorly understood. We believe that this is an especially important contribution to the field given the relatively large number of newly proposed methods that might overstate the effectiveness of these approaches by omitting larger corpora often found in the information retrieval community. In addition, we find it compelling that even without novel methods, our rigorous ablation is able to produce SOTA results on established tasks (Table 2), outperforming more complicated approaches such as GenRet (Sun et al., 2023).
>
> 2. Clarification on statements: We apologize if some statements were unclear, and will update the paper to clarify.
> In Line 123, "implicitly" is used generally to refer to methods that expand the parameter count beyond what is initially used in the original T5 model’s pretraining checkpoint. For example, even though initialized at T5-Base, NCI’s PAWA decoder expands the original 220M parameters to 700M+. This fact is not clearly stated in Wang et al. 2022 and models are still referenced by their pretrained size “T5-Base”, and thus we consider it “implicit”. The same goes with the Atomic ID setting where adding more documents to the same base T5 configuration implicitly results in more parameters, with no direct changes to the architecture.
> An example of the “common misconception” we refer to in Line 140 can be found in Section 4.4 in the Neural Corpus Indexer paper (Wang et al. 2022), “This implies that the model capacity has a critical impact on the retrieval performance …” and “For a larger corpus, one may need to increase the model size to obtain satisfactory performance.” Through our experiments, it is clear that smaller models (T5-Base) would find it hard to properly handle large corpora. However, just scaling from T5-XL to XXL is not sufficient in enhancing downstream effectiveness, and more advancements in modeling are required to meet the effectiveness of SOTA dual encoders. We will clarify these points in the revised version.
>
> 3. Evaluation Metrics: We understand your concern about the use of a single metric for each dataset. In our study, we chose the official metric for each dataset we investigated, unlike some of the prior work in generative retrieval evaluation using some non-traditional IR metrics. However, we also agree that including additional metrics could provide a more comprehensive evaluation. The following table expands on the top-scoring methods in our paper from Table 4 (MSMarcoFULL), and also adds two additional sets widely considered in the IR community (TREC DL19/20):
>
> | Model                              | Params | MRR@10 | Recall@100 | NDCG@10 | NDCG@100 | TREC DL 19 NDCG@10 | TREC DL 19 NDCG@100 | TREC DL 20 NDCG@10 | TREC DL 20 NDCG@100 |
> |------------------------------------|--------|--------|------------|---------|----------|--------------------|---------------------|--------------------|---------------------|
> | Base D2Q Only + Atomic ID          | 7.0B   |   24.2       | 80.47   | 29.73    | 36.35              | 49.75               | 50.21              | 53.75               | 56.01 |
> | Large D2Q Only + Naive ID          | 783M   |   21.4       | 82.03   | 27.26    | 34.32              | 49.92               | 46.1               | 45.16               | 49.18 |
> | Large D2Q Only + PAWA (2D Sem.)    | 2.1B   |     19.8       | 73.21   | 24.79    | 31.3               | 46.57               | 45.39              | 50.13               | 50.56 |
> | XL D2Q Only + Naive ID             | 2.8B   |    26.7       | 84.74   | 33.17    | 39.22              | 54.95               | 51.63              | 52.19               | 54.07 |
> | XXL D2Q Only + Naive ID            | 11B    |   24.3       | 81.87   | 30.56    | 36.96              | 52.05               | 47.67              | 49.03               | 52.1 |
>
> These additional densely judged evaluation datasets, as opposed to the sparse judgments in the MS MARCO v1 Passage Ranking Task, seek to address some of the issues with the original task. We see that our original findings generally hold under these new metrics, and believe that this addition reaffirms the soundness of our study.
>
> For these top-5 Table 4 runs we additionally have {Recall, NDCG, Precision}@{1, 5, 10, 20, 50, 100} and MRR@100. Additionally, for runs in Table 3, we also have Recall@{1, 5, 10, 20}, NDCG@10, and Precision@{1, 5, 10, 20} values. We do not include them here due to the sheer size of the tables, but we plan to include them in the Appendix for our Camera Ready version. If there is a particular comparison you’re interested in on a certain metric we’d be happy to accommodate it during the discussion period. We hope that this further reinforces our findings for the community, given we are the first to tackle this task at this scale with extensive experimentation.
>
> 4. Terminology and Abbreviations: We apologize for any confusion caused by the use of specific names and abbreviations. In the revised version, we will ensure that all terms and abbreviations are clearly defined when first introduced.
>
> 5. Related works: As described in our limitations section, we intentionally omit the study of natural language and learned document identifiers from this work. Given the wide search space and the numerous experiments we already present, we believe this is best suited for future work that can rigorously dissect the relationship between retrieval performance and extractive (e.g. Bevilacqua et al., 2022) or abstractive (generated) document identifiers, natural language tokens or learned tokens, and the quality of the underlying docid generation model.
>
> In addition, the missing references provided seem to be published in proceedings that appeared after the submission of this paper. Regardless, we'd be happy to include them in our related works in the final version.
>
> Thanks again for your time and thoughts! We believe that these changes and additions will significantly improve the paper.

---

### Official Review · Reviewer_vuyp · 2023-08-12

**Soundness:** 4

**Excitement:**

4: Strong: This paper deepens the understanding of some phenomenon or lowers the barriers to an existing research direction.

**Paper Topic And Main Contributions:**

The authors experiment with the generative retrieval approach in scale. The paper contains an extensive evaluation across multiple experiments, along with the authors' findings. First, the authors evaluate several existing methods on multiple datasets; then they propose their own method and show its strong performance in small scale; and finally they scale up the corpus size from 100k to several millions of passages.

**Reasons To Accept:**

- Extensive experiments
- Findings that can be useful for the IR community for future research

**Reasons To Reject:**

- For a complete evaluation, I'd prefer to see more than a single metric for each dataset.
- The paper can be hard to follow in some parts: It is full of specific names and abbreviations; moreover, some of them are not defined, such as in the results tables  (e.g., At, Nv) - please define and fully describe them. See below for more.

**Reproducibility:**

3: Could reproduce the results with some difficulty. The settings of parameters are underspecified or subjectively determined; the training/evaluation data are not widely available.

**Reviewer Confidence:**

2: Willing to defend my evaluation, but it is fairly likely that I missed some details, didn't understand some central points, or can't be sure about the novelty of the work.

**Typos Grammar Style And Presentation Improvements:**

- The term "docid" is referred to in the introduction, before its definition in Section 2.1.

---

> ### Author Rebuttal · Authors · 2023-08-28
>
> We appreciate your time and effort in reviewing our paper and are grateful for your constructive feedback. We hope the following can help address your concerns:
>
> 1. Evaluation Metrics: We understand your concern about the use of a single metric for each dataset. In our study, we chose the official metric for each dataset we investigated, unlike some of the prior work in generative retrieval evaluation using some non-traditional IR metrics. However, we also agree that including additional metrics could provide a more comprehensive evaluation. The following table expands on the top-scoring methods in our paper from Table 4 (MSMarcoFULL), and also adds two additional sets widely considered in the IR community (TREC DL19/20):
>
> | Model                              | Params | MRR@10 | Recall@100 | NDCG@10 | NDCG@100 | TREC DL 19 NDCG@10 | TREC DL 19 NDCG@100 | TREC DL 20 NDCG@10 | TREC DL 20 NDCG@100 |
> |------------------------------------|--------|--------|------------|---------|----------|--------------------|---------------------|--------------------|---------------------|
> | Base D2Q Only + Atomic ID          | 7.0B   |   24.2       | 80.47   | 29.73    | 36.35              | 49.75               | 50.21              | 53.75               | 56.01 |
> | Large D2Q Only + Naive ID          | 783M   |   21.4       | 82.03   | 27.26    | 34.32              | 49.92               | 46.1               | 45.16               | 49.18 |
> | Large D2Q Only + PAWA (2D Sem.)    | 2.1B   |     19.8       | 73.21   | 24.79    | 31.3               | 46.57               | 45.39              | 50.13               | 50.56 |
> | XL D2Q Only + Naive ID             | 2.8B   |    26.7       | 84.74   | 33.17    | 39.22              | 54.95               | 51.63              | 52.19               | 54.07 |
> | XXL D2Q Only + Naive ID            | 11B    |   24.3       | 81.87   | 30.56    | 36.96              | 52.05               | 47.67              | 49.03               | 52.1 |
>
>
> These additional densely judged evaluation datasets, as opposed to the sparse judgments in the MS MARCO v1 Passage Ranking Task, seek to address some of the issues with the original task. We see that our original findings generally hold under these new metrics, and believe that this addition reaffirms the soundness of our study.
>
> For these top-5 Table 4 runs we additionally have {Recall, NDCG, Precision}@{1, 5, 10, 20, 50, 100} and MRR@100. Additionally, for runs in Table 3, we also have Recall@{1, 5, 10, 20}, NDCG@10, and Precision@{1, 5, 10, 20} values. We do not include them here due to the sheer size of the tables, but we plan to include them in the Appendix for our Camera Ready version. If there is a particular comparison you’re interested in on a certain metric we’d be happy to accommodate it during the discussion period. We hope that this further reinforces our findings for the community, given we are the first to tackle this task at this scale with extensive experimentation.
>
> 2. Terminology and Abbreviations: Thank you for your feedback. In the revised version, we will ensure that all terms and abbreviations are clearly defined when first introduced.
>
> We believe these revisions will address your concerns and significantly improve the quality of our paper!

---

### Meta-Review · Area_Chair_zu7w · 2023-09-18

**Recommendation:** 4

**Metareview:**

This paper is an empirical study of generative retrieval techniques, carried out at a scale that is beyond what has been previously studied. The overall finding is that generative techniques are effective at smaller scales, but more research is needed for scaling to much larger settings.

Reviewers found this paper to be well-written, a useful empirical study, and likely of value to parts of the community ("could be valuable for the researchers in the field").

The main concerns among reviewers had to do with the diversity of results, as the primary focus was on a single metric on a single large scale dataset. In their rebuttal, the authors have helpfully provided additional results which include additional metrics and additional datasets, and this seems to have allayed the first concern at least. Nevertheless, two reviewers remained ambivalent in terms of excitement, even after taking the rebuttal into consideration. Unfortunately, one of these reviewer was somewhat confused about whether new experimental results could be considered as part of the rebuttal. The policy this year is a departure from past years, so it is understandable that they thought such revisions might be off limits, when in fact new results were within the scope for evaluation.

All reviewers agreed that this paper is worthy of publication. However, the one reviewer who was by far the most confident, who also engaged with the authors during the rebuttal, remained unconvinced to increase their excitement score above ambivalent. On the one hand, they suggest that "this paper will be an excellent and comprehensive empirical study that will benefit the research field of generative retrieval". On the other, they remain concerned that the paper is limited to generative retrieval based on only virtual tokens, and as such, "the impact of this paper is still limited", and they have not been convinced to raise their scores above 3.

Ultimately this paper is somewhat difficult to place. There seems to be consensus that this is a paper that will be interesting and useful to those who work on generative retrieval, despite being primarily an empirical study. That being said, the authors were unable to convince two reviewers to increase their scores, despite providing new results that seem to be precisely what these reviewers were seeking.

---

### Decision · Program_Chairs · 2023-10-07

**Decision:**

Accept-Main

**Comment:**

This paper is an empirical study of generative retrieval techniques, carried out at a scale that is beyond what has been previously studied. The overall finding is that generative techniques are effective at smaller scales, but more research is needed for scaling to much larger settings.

Reviewers found this paper to be well-written, a useful empirical study, and likely of value to parts of the community ("could be valuable for the researchers in the field").

The main concerns among reviewers had to do with the diversity of results, as the primary focus was on a single metric on a single large scale dataset. In their rebuttal, the authors have helpfully provided additional results which include additional metrics and additional datasets, and this seems to have allayed the first concern at least. Nevertheless, two reviewers remained ambivalent in terms of excitement, even after taking the rebuttal into consideration. Unfortunately, one of these reviewer was somewhat confused about whether new experimental results could be considered as part of the rebuttal. The policy this year is a departure from past years, so it is understandable that they thought such revisions might be off limits, when in fact new results were within the scope for evaluation.

All reviewers agreed that this paper is worthy of publication. However, the one reviewer who was by far the most confident, who also engaged with the authors during the rebuttal, remained unconvinced to increase their excitement score above ambivalent. On the one hand, they suggest that "this paper will be an excellent and comprehensive empirical study that will benefit the research field of generative retrieval". On the other, they remain concerned that the paper is limited to generative retrieval based on only virtual tokens, and as such, "the impact of this paper is still limited", and they have not been convinced to raise their scores above 3.

Ultimately this paper is somewhat difficult to place. There seems to be consensus that this is a paper that will be interesting and useful to those who work on generative retrieval, despite being primarily an empirical study. That being said, the authors were unable to convince two reviewers to increase their scores, despite providing new results that seem to be precisely what these reviewers were seeking.